# Environmentally Friendly Polyvinyl Alcohol−Alginate/Bentonite Semi-Interpenetrating Polymer Network Nanocomposite Hydrogel Beads as an Efficient Adsorbent for the Removal of Methylene Blue from Aqueous Solution

**DOI:** 10.3390/polym13224000

**Published:** 2021-11-19

**Authors:** Mona A. Aziz Aljar, Suad Rashdan, Ahmed Abd El-Fattah

**Affiliations:** 1Department of Chemistry, College of Science, University of Bahrain, Sakhir P.O. Box 32038, Bahrain; maljar@uob.edu.bh (M.A.A.A.); srashdan@uob.edu.bh (S.R.); 2Department of Materials Science, Institute of Graduate Studies and Research, Alexandria University, Alexandria 21526, Egypt

**Keywords:** alginate, polyvinyl alcohol, bentonite clay, nanocomposites hydrogel, cationic dyes, water remediation

## Abstract

Hazardous chemicals like toxic organic dyes are very harmful to the environment and their removal is quite challenging. Therefore there is a necessity to develop techniques, which are environment friendly, cost-effective and easily available in nature for water purification and remediation. The present research work is focused on the development` and characterization of the ecofriendly semi-interpenetrating polymer network (semi-IPN) nanocomposite hydrogels composed of polyvinyl alcohol (PVA) and alginate (Alg) hydrogel beads incorporating natural bentonite (Bent) clay as a beneficial adsorbent for the removal of toxic methylene blue (MB) from aqueous solution. PVA−Alg/Bent nanocomposite hydrogel beads with different Bent content (0, 10, 20, and 30 wt%) were synthesized via external ionic gelation method. The designed porous and steady structure beads were characterized by the use of Fourier transform infrared spectroscopy (FTIR), energy-dispersive X-ray spectroscopy (EDX), and scanning electron microscopy (SEM). The performance of the beads as MB adsorbents was investigated by treating aqueous solutions in batch mode. The experimental results indicated that the incorporation of Bent (30 wt%) in the nanocomposite formulation sustained the porous structure, preserved water uptake, and increased MB removal efficiency by 230% compared to empty beads. Designed beads possessed higher affinity to MB at high pH 8, 30 °C, and fitted well to pseudo-second-order kinetic model with a high correlation coefficient. Moreover, the designed beads had good stability and reusability as they exhibited excellent removal efficiency (90%) after six consecutive adsorption-desorption cycles. The adsorption process was found be combination of both monolayer adsorption on homogeneous surface and multilayer adsorption on heterogeneous surface. The maximum adsorption capacity of the designed beads system as calculated by Langmuir isotherm was found to be 51.34 mg/g, which is in good agreement with the reported clay-related adsorbents. The designed semi-IPN PVA−Alg/Bent nanocomposite hydrogel beads demonstrated good adsorbent properties and could be potentially used for MB removal from polluted water.

## 1. Introduction

Methylene blue (MB) is a well-known cationic phenothiazine dye, with versatile industrial applications in the textile, leather, plastics, paper and cosmetics industry and also used as a biological stain and photosensitizer in photodynamic therapy [1,2,3,4,5,6,7]. However, MB is nonbiodegradable due to its complex aromatic structure, contributing to a prolonged toxicity [8,9,10]. Consequently, it is necessary to remove toxic MB dye from industrial effluents and wastewater for a more secure and sustainable environment [11,12].

Various conventional physicochemical techniques have been proposed and introduced for the purification of toxic dyes from water from including photocatalysis, chemical oxidation, ozonation, membrane filtration, coagulation and flocculation, as well as adsorption [13,14,15,16,17]. Among all of these techniques, the adsorption method is easily employed, fast and highly efficient [18,19,20]. Numerous different adsorbents have been synthesized such as activated carbons [21,22], porous ceramics [23,24], metal oxides [25], and polymeric hydrogels [26,27,28,29,30] for the remediation of contaminated water. Indeed, from both an environmental and industrial point of view, an ideal adsorbent should have large surface area, porous structure, available active functional sites, high mechanical stability and strength, as well as low cost and be eco-friendly [31,32]. 

In recent years, semi-interpenetrating polymer network (semi-IPN) biodegradable hydrogels consisting of linear polymer chains of one component entangled with other crosslinked polymer network have attracted wide attention as adsorbents for removal of pollutants from industrial water [33]. This is due to their desirable properties such as, biodegradability, biocompatibility, high elasticity and good mechanical strength [34]. In addition, during the removal process, the dye molecules can rapidly penetrate into the porous semi-IPN structures in solution and therefore, combine with the hydrophilic active site groups like hydroxyl (−OH) and carboxyl (−COOH) present on their surfaces [26,35].

A striking candidate that constitutes the most investigated matrix in the preparation of semi-IPN hydrogels is sodium alginate (Alg), a naturally occurring anionic biopolymer obtained from brown algae. It has unique properties, including low cost, biodegradability, biocompatibility, and flexibility in hydrated environments. In addition, the ionized negatively-charged carboxylate (−COO^−^) groups of Alg are able to attract cationic MB from aqueous solutions [36,37,38]. Blending polyvinyl alcohol (PVA), which possesses high elasticity, and mechanical strength forms a semi-IPN hydrogel structure with an Alg hydrogel matrix with outstanding mechanical performance [39]. Despite that fact, several researchers have reported that semi-IPN PVA−Alg hydrogel has limited application in dye removal from wastewater owing to its relatively low adsorption capacity and selectivity [37,38,40]. 

Nanoparticles based on clay minerals, have received extensive attention due to their remarkably low-cost, large abundance, high specific surface area, and good intrinsic adsorption characteristics [11,23,31]. Because of the anionic nature of the clay nanoparticles in aqueous media, a number of studies have been reported the efficacy of clay minerals for removing cationic dyes from wastewater [41,42]. Bentonite (Bent) is a naturally occurring clay that is composed tetrahedral Si layer and octahedral Al layer which are interlinked and contain reactive –OH groups. The charged layers are neutralized by the presence of cations like Na^+^ and K^+^ and the interlayer space holding some amount of water [43]. Owing to these exciting features, incorporating Bent nanoparticles into the semi-IPN PVA−Alg hydrogel matrices can significantly improve adsorption performance and further increase the mechanical stability of the hydrogel adsorbents [44,45].

The objective of the present research is to continue to explore the advantages of both ecofriendly polymeric hydrogel and natural nanoparticle-based materials. In particular, our aim is to synthesize and characterize biodegradable nanocomposite hydrogel, with enhanced mechanical stability and adsorption properties as well as understanding the adsorption mechanism in order to assess the best commercial formulations for industrial applications. 

In this work, semi-IPN PVA−Alg/Bent nanocomposite hydrogel beads were prepared via external ionic gelation method and applied to the removal of toxic MB dye from aqueous solution. The structure, composition, morphology, and swelling of the assembled beads were investigated. Many factors affecting the MB adsorption from the aqueous solution by PVA−Alg/Bent nanocomposite hydrogel beads were studied in detail, including different Bent contents, pH values, contact time, initial MB concentration, different adsorbent dosage, temperature, and reusability. Further, adsorption kinetics, adsorption isotherms, and adsorption mechanism were analyzed.

## 2. Materials and Methods or Experimental

### 2.1. Materials

Polyvinyl alcohol (PVA; molecular weight: 13,000−23,000 g/mol; 89 mol% hydrolyzed), low viscosity alginate (Alg) powder, calcium chloride (CaCl_2_), hydrochloric acid (HCl), sodium hydroxide (NaOH), and methylene blue (MB; chemical formula: C_16_H_18_N_3_SCl, molecular weight: 319.85 g/mol, solubility in water: 40 g/L, λ_max_ = 663 nm, as shown in Figure 1a) were purchased from Sigma–Aldrich Co. (Munich, Germany) and used without further purification. Bentonite (Bent) clay nanoparticles (Bent; molecular weight: 180.1 g/mol; chemical formula: H_2_Al_2_O_6_Si;), was obtained as a gift from Sphinx Milling Station Co. (Alexandria, Egypt). As shown in Figure 1b, an SEM image of Bent clay revealed a spherical nanosized particle morphology with average particle size of 26 ± 1.43 nm and narrow size distribution ranging from 18 ± 1.72 nm to 37 ± 53 nm as determined by SEM image analysis software. The deionized water (0.055 µS/cm) was used in all experiments.

### 2.2. Preparation of Semi-IPN Nanocomposite Hydrogel Beads

PVA (2 g) and Alg (2 g) were dissolved in deionized water (100 mL) in a water bath at 90 °C under constant stirring at 350 rpm for 2 h to achieve a homogeneous solution. After complete dissolution, the solution was stirred for an additional 1 h and cooled to room temperature. Then, a predetermined content of the Bent powder (10, 20 and 30 wt% in relation to the polymers) was dispersed in the aqueous PVA−Alg solution via sonication at 40 °C for 30 min, and the mixture was vigorously stirred for 1 h in a shaker. The obtained viscous suspension was slowly added dropwise by a syringe into the stirring 3% (*w*/*v*) CaCl_2_ solution to form semi-IPN nanocomposite hydrogel beads. To complete the gelation reaction, the beads were maintained in the CaCl_2_ solution overnight. Finally, the beads were filtered, washed three times with deionized water to remove any unreacted CaCl_2_, dried for 48 h at room temperature, and stored in a clean glass bottle for further characterization and adsorption study. Control hydrogel beads without Bent clay were prepared using the same previous steps.

### 2.3. Characterization Methods

#### 2.3.1. Fourier Transform Infrared Spectroscopy (FTIR)

The chemical composition of the Bent, PVA‒Alg control hydrogel and PVA‒Alg/Bent nanocomposite hydrogel were analyzed by FTIR spectroscopy using a Spectrum Two FT-IR spectrometer (Perkin-Elmer, Waltham, MA, USA). The spectra were recorded throughout the wavenumber range from 4000 to 400 cm^−1^, at a resolution of 4 cm−1 with 32 scans per sample. The sample was prepared using the KBr disc technique in which the sample (5 mg) was compressed with KBr powder (200 mg) by hydraulic pressure to form a disk with a diameter of 12 mm.

#### 2.3.2. Scanning Electron Microscopy (SEM)

A scanning electron microscope (SEM) was used to assess the surface morphology of the desired beads. For this purpose, a JSM-5300 instrument (JEOL, Tokyo, Japan) was used, which was operated at 20 kV. Prior to SEM imaging, the samples were ultrasonically washed for 30 s and sputter-coated with gold to a thickness of 0.04 µm in a sputter-coating unit (JFC 1100 E, JEOL).

#### 2.3.3. Energy-Dispersive X-ray Spectroscopy (EDX)

Elemental composition of the Bent, PVA‒Alg control hydrogel and PVA‒Alg/Bent nanocomposite hydrogel was determined using an energy dispersive X-ray (EDX) microanalysis attachment on the SEM system. Analysis was performed on uncoated samples at 15 kV for 60 s.

#### 2.3.4. Swelling Behavior

To estimate the swelling behavior of the beads, the dried beads were immersed in deionized water (500 mL) at different pH values (3, 7, and 9) for 5 h at room temperature until the beads reached equilibrium swelling. During this experiment, water was occasionally exchanged several times. The swollen beads were taken out of the aqueous solution at fixed time scheduled, surface wiped and weighted to obtain their wet weights (W_w_). Then they were dried at 40 °C in an oven until constant weight was achieved. After which, they were weighted again to determine their dry weights (W_d_). The average of five measurements was taken as the final result. The percentage of swelling ratio by each sample was calculated according to Equation (1) [46]:Swelling ratio (%) = [(W_w_ − W_d_)/W_d_] × 100(1)

### 2.4. MB Adsorption Experiments

The adsorption of MB from aqueous solution onto PVA−Alg/Bent nanocomposite hydrogel beads was systematically estimated. Before the adsorption experiments, a stock solution of MB (500 ppm) was prepared by dissolving an accurately weighed amount of MB powder in deionized water. The prepared stock solution was diluted to various MB concentrations of the same pH and absorbance was measured by using a UV/Vis spectrophotometer (Lambda 35, Perkin-Elmer) which recorded the characteristic absorption peak of MB at wavelength of 663 nm. A calibration curve was constructed with reference to absorbance values of corresponding concentration of MB. The calibration curve equation was used to find the concentration of MB before and after adsorption. The adsorption experiments were studied by batch adsorption technique in which a certain amount of dried beads was added to a 100 mL Erlenmeyer flask containing MB solution (50 mL) of a certain concentration and placed in a mechanical water bath shaker at 150 rpm for a predetermined period of time. Then, the supernatant was centrifuged and the concentration was measured by UV/Vis spectrophotometer. Effect of different Bent content (10, 20, and 30 wt%), different pH (2−10), contact time (20−320 min), initial MB concentration (50−350 ppm), PVA−Alg/Bent dosage (0.1−2 g), and temperature (20, 30, 40, and 50 °C) have been explored to evaluate the performance of adsorbent beads. The equilibrium adsorption capacity, qe, (mg/g) of MB onto adsorbent beads as well as the removal percentage (%) were calculated from Equation (2) [47] and Equation (3) [48,49] respectively: q_e_ = (C_o_−C_e_) × V/m(2)
Removal (%) = [(C_o_−C_e_)/ C_o_] × 100(3)
where C_o_ and C_e_ are the initial and equilibrium concentrations of MB (mg/L), respectively, m is the adsorbent mass (g) and V is the MB solution volume (L).

### 2.5. Adsorbent Reusability Study

The reusability of semi-IPN PVA−Alg/Bent adsorbent beads was determined in a repeated adsorption−desorption process. Adsorbent beads (1.5 g) were added to MB solution (200 mg/L, 50 mL) at pH 8. The mixture was shaken in an orbital shaker incubator for 300 min at 30 °C. The saturated MB-loaded beads were collected out, washed thoroughly with deionized water, and dried for 48 h at room temperature. For the regeneration process, the dried beads were immersed in HCl solution (0.1 M, 50 mL) and shaken at 150 rpm for 30 min, then separated from the solution, rinsed multiple times by deionized water till no more MB leached out, and finally dried for 48 h at room temperature for reuse. The regenerated beads were reused to conduct the same adsorption procedure at the same conditions. Six adsorption–desorption cycle tests, were carried out.

### 2.6. Adsorbent Isotherms

The adsorption mechanisms were studied by Freundlich and Langmuir isotherm models. The Freundlich isotherm adsorption formula is expressed in Equation (4) [50,51].
(4)$$\ln\;q_{e}=\ln\;K_{F}+\frac{1}{n\;}\;\ln C_{e}\;$$
where *K_F_* [(mg/g)(L/mg)^1/n^] and *n* are the Freundlich constants, which represent the adsorption capacity and adsorption intensity, respectively. For favorable and multilayer cooperative adsorption, 1/*n* value should be between 0 and 1 [51].

The Langmuir isotherm adsorption Equations (5) and (6) [50,51] were defined as follows:(5)$$\frac{C_{e}}{q_{e}}=\frac{1}{K_{L}q_{m}}+\frac{C_{e}}{q_{m}}\;\;$$
(6)$$R_{L}=\frac{1}{1+K_{L}C_{o}}$$
where *q_m_* (mg/g) is the maximum adsorption capacity, *K_L_* (L/mg) is the Langmuir constant related to the affinity of the binding sites, and *R_L_* is the constant separation factor. R_L_ value between 0 and 1 states a favorable adsorption process [50].

### 2.7. Adsorbent Kinetics

Both pseudo-first and pseudo-second order kinetic models were adopted to estimate the adsorption kinetics of MB dye onto PVA−Alg/Bent nanocomposite hydrogel beads. The corresponding Equations (7) and (8) [50,51] displayed in their linear forms are as follows:(7)$$\ln\left(q_{e}\;-\;q_{t}\right)\;=\ln q_{e}\;-\;K_{1}t$$
(8)$$\frac{t}{q_{t}}=\frac{1}{K_{2}q_{e}^{2}}\;+\frac{1}{q_{e}}t\;\;$$
where *q_e_* and *q_t_* (mg/g) are the amounts of MB adsorbed at equilibrium and at time *t*, respectively. *K*_1_ (1/min) and *K*_2_ (g/mg.min) are the rate constants of pseudo-first order and pseudo-second-order kinetics, respectively.

### 2.8. Adsorbent Kinetics

The significant differences between the data among the experimental groups were analyzed using a one-way ANOVA test followed by a Tukey’s post-hoc test and a Student *t*-test. All data were represented as mean ± standard deviation (M ± SD) and each experiment being performed in at least five replicates. A *p* value < 0.05 was considered significant.

## 3. Results and Discussion

### 3.1. Design Rationale of the Nanocomposite Hydrogel Beads

The semi-IPN PVA‒Alg/Bent nanocomposite hydrogel beads were successfully prepared by external ionic gelation method with Ca^2+^ ions as crosslinkers [44,47]. The digital images of the wet and dry beads before and after adsorption of MB are displayed in Figure 2a,b,c. As shown in Figure 2a, the incorporation of Bent nanoparticles into the hydrogel beads was confirmed by the apparent beige color of the beads. The wet beads exhibited uniformly spherical shapes with a smooth surface and had a size of approximately 4 mm (Figure 2a). However, during air drying, the initial spherical shape of the beads was slightly changed, resulting in a rough surface morphology without a marked collapse on the surface, indicating an improved mechanical stability (Figure 2b). Indeed, the deformation in shape is inevitable when water was evaporated from the wet hydrogel beads throughout the drying process, causing volume shrinkage of hydrogel beads [52]. Figure 2c reveals black color of the beads indicated encapsulation of MB in the nanocomposite hydrogel beads.

In this work, the PVA: Alg weight ratio and the content of Bent nanoparticles were fixed at (1:1) and 30 wt%, respectively. These values were selected because they represent an optimum performance of efficient adsorptive removal of MB dye and mechanical stability of the hydrogel beads [44]. To assess the adsorption ability of the designed nanocomposite hydrogel beads, preliminary adsorption tests were performed. As shown in Figure 2d, the control hydrogel beads had limited removal rate (28.53%) for MB. Nevertheless, the removal rate of MB was significantly improved by increasing the content of Bent nanoparticles in the hydrogel beads. More specifically, the incorporation of 10 wt%, 20 wt%, and 30 wt% Bent nanoparticles in the hydrogel beads increased the removal rate of MB dye by 165%, 190%, and 230%, respectively, compared to the control hydrogel beads. However, Bent contents exceeding the selected value were not used to avoid aggregation of Bent nanoparticles, which in turn may deteriorate the overall mechanical stability and beads strength [53,54]. After taking the factors of adsorption amount and mechanical strength into consideration, semi-IPN PVA–Alg/Bent (30 wt%) nanocomposite hydrogel with appropriate MB removal percentage (94.64%) and stiffness was selected for subsequent characterizations and adsorption study.

### 3.2. Structure and Chemical Composition

Figure 3 displays the FTIR spectra of the pristine Bent, semi-IPN PVA–Alg control hydrogel, and semi-IPN PVA–Alg/Bent nanocomposite hydrogel. Bent exhibits characteristic absorption peaks at 1054, 982, 3674, and 835 cm^−1^. The absorption peak at 1054 cm^−1^ is attributed to the stretching vibrations of Si–O–Si group, whereas the peak at 950 cm^−1^ is related to Al–OH–Al bending vibrations [50,53]. Besides, very weak bands at 3674 and 835 cm^−1^ are corresponding to O–H and Si–O stretching vibrations of silanol group of bent. The spectrum of the PVA–Alg hydrogel, shows a broad overlapped absorption band in the region of 3300−3400 cm^−1^ which is related the stretching vibration of O–H of the PVA and Alg [44,47]. Absorption peaks in the area of 1635 cm^−1^ and 1428 cm^−1^ are due to asymmetric and symmetrical stretching vibrations of the –COO of Alg [44,55]. The peak appeared at 2947 cm^−1^ was the typical stretching vibration for C–H in PVA. All the characteristic peaks of PVA, Alg, and Bent were observed in the FTIR spectrum of the nanocomposite hydrogel. In addition, the small shift in the position of the band near 1300 cm^−1^ would be related to the presence of hydrogen bonds between O–H groups of PVA and Alg or with Si-O groups of Bent [44,53,55]. These results suggested that the Bent clay had successfully incorporated in the PVA–Alg hydrogel polymer matrix.

The presence of Bent nanoparticles homogeneously distributed in the hydrogel matrix was further confirmed by EDX elemental analysis as shown in Figure 4. EDX spectrum of Bent clay (Figure 5a) confirmed the presence of its main constituents such as O (51.70%), Si (21.89%), Al (9.81%), Fe (6.88%), and C (4.61%) [50,56]. The main characteristic peaks attributed to C (40.95%) and O (52.19%) elements are detected in the EDX spectrum of control hydrogel (Figure 4b), which are due to the contribution of PVA and Alg component. It was worth noting that the peaks of Si, Al and Fe elements can be clearly observed in the EDX spectrum (Figure 4c) of PVA–Alg/Bent nanocomposite hydrogel, indicated the successful incorporation of Bent nanoparticles on the hydrogel. 

### 3.3. Morphological Observation 

Figure 6 presents the SEM micrographs of the dried entire beads and their surfaces for both control and nanocomposite hydrogels. SEM micrographs revealed a slight increase in the size of the nanocomposite bead (Figure 5a) compared to the unfilled bead (Figure 5c). Besides, both beads exhibited an irregular gully shape with some cracks and pores on the surface. However, as displayed in Figure 5b,d, the incorporation of Bent (30 wt%) in the hydrogel resulted in a clear decrease in the size of the pores and the thickness of the cracks. Furthermore, the nanocomposite hydrogel showed uniform dispersion of the clay nanoparticles within the polymer matrix, leaving a relatively rough surface (Figure 5d). This structure will provide more contact sites and convenient diffusion channels, which may contribute to a better MB dye adsorption by PVA–Alg/Bent beads [44,52,56].

### 3.4. Swelling Behavior

The swelling behavior of the semi-IPN PVA–Alg/Bent (30 wt%) nanocomposite hydrogel beads in aqueous solution at different pH values is depicted in Figure 6. Clearly, all beads showed a similar tendency with different degrees of hydration. As can be seen from Figure 6a, the swelling ratio enhanced slightly during the initial swelling stage (20–150 min), and then turn into slower until reaching a plateau (150–320 min). One reasonable explanation for this observation was attributed to the presence of –OH groups on the polymer chains and clay surfaces that attract water molecules from the soaking medium, promoting their penetration into the gel-network, and eventually, elevating water uptake ability of the nanocomposite beads [56,57]. Moreover, hydrogel beads exhibited obvious pH-dependent swelling behavior, where the water uptake of the beads increased significantly with increasing pH of the solution. More specifically, at 150 min, the swelling rate increased gently within pH 3−7 (from 25.5–30.2%) and increased speedily in the range of pH 7−9 (from 30.2–40.6%) indicating that the structure of the cross-linked hydrogel changed at different pH values (Figure 6b). The low swelling capacity at acidic solution (pH 3) can be due to the protonation of –COO^−^ to –COOH groups of the hydrogel matrix. Consequently, the beads existed in collapsed state via the hydrophobic interaction, lead to more rigid gel-networks structure, which restricted traversing of water [44,51,57]. In neutral and alkaline solution (pH 9), some of the –COOH groups in the hydrogel ionized to –COO^−^, which establishing repulsions between the negatively charged chains, facilitating the expansion of hydrogel and thus enhancing the swelling rate [51,57].

### 3.5. Adsorption Studies of MB

To study the suitability of the designed semi-IPN PVA–Alg/Bent nanocomposite hydrogel beads as adsorbent for MB dye, adsorption experiments were first conducted utilizing a batch experimental setup. The influences of various parameters on adsorption behaviors of the beads, such as pH, contact time, initial dye concentration, beads dosage and temperature, were explored.

#### 3.5.1. Effect of Solution pH

The pH of dye solution is the most vital parameter controlling the efficiency of dye adsorption process [57]. The effect of solution pH on the adsorption capacity and removal percentage of MB dye by the nanocomposite hydrogel beads is illustrated in Figure 7a. As the pH increased from 2 to 5, the adsorption capacity of MB was low and kept roughly constant, however, raised significantly within the pH range from 5 to 8, and finally decreased at pH values greater than 8. The maximum percentage of dye removal (94.64%) was achieved at pH 8. As mentioned above, the pH of the dye solution may affect the ionization of functional groups on the designed nanocomposite hydrogel adsorbent, which worked as the active sites of the hydrogel adsorbent, in turn, governing the electrostatic attraction between the anions on the hydrogel matrix and the cation on the MB dye [51,53,58]. At acid conditions, the negative surface charges of the nanocomposite hydrogel adsorbent were reduced, which were undesirable for the removal of positively charged MB dye. At alkaline pH 8, the surface negative charges of the hydrogel increased, which was favorable for the removal of MB due to the enhanced electrostatic attraction to the positively charged MB. Above pH 8, the surface of the hydrogel exhibited negative electricity, which lessening the electrostatic attraction between MB and the hydrogel, consequently, causing smaller removal values [55,58].

#### 3.5.2. Effect of Contact Time

Contact time is a significant factor that indicates whether the adsorbent removes the target contaminant to reach the equilibrium [55]. The variation of MO removal rate and adsorption capacity over contact time were shown in Figure 7b. It was found that the adsorption capacity, as well as removal percentage, were improved with increasing immersion contact time. A fast adsorption for the initial 60 min and a relatively slower adsorption after 60 min till 5 h were observed. At the same time, the color of the hydrogels became dark gradually, whereas the MB solution turned clear from an initial blue to completely transparent as shown in the inset of Figure 7b. These may be attributed to the filling of all the surface binding sites and diffusion of the MB molecules into pores of the designed beads. At the starting of adsorption process, there were plentiful active sites on the surface of the beads, which could combine with the MO molecules once they were added into dye solution [57,58]. Moreover, the high MB concentration in the solution provided the apparent driving force for MB adsorption supported by concentration gradient. In the later stage of the adsorption process, where the concentration of MB decreased, the adsorption sites on the surface of the beads tended to be saturated. Therefore, the diffusion and adsorption of MB molecules were reduced and hence the adsorption rate decreased continuously [59].

#### 3.5.3. Effect of Initial MB Concentration

In batch adsorption, the initial concentration of the MB molecules is the main driving force offsetting the mass diffusion resistance force [56]. Figure 7c shows the influence of the initial concentration of MB on adsorption capacity and removal percentage of MB. It could be noticed that the dye adsorption capacity increased with increasing initial MB concentration while the removal percentage declined. For instance, when the initial concentration of MB was 50, 200 and 300 mg/L, the removal rate of beads was about 97, 85 and 69%, respectively. This indicated that at a lower concentration of MB, nanocomposite hydrogel beads had an abundance of adsorption active sites leading to a high adsorption for the majority of MB molecules [60]. However, saturation of adsorption sites would be established with further increases in the MB concentration, and adsorption became difficult since the increased numbers of MO molecules attached to the surface of the beads would repel free MO molecules [58,59,60].

#### 3.5.4. Effect of Adsorbent Dosage

Adsorbent dosage is another key factor in the dye adsorption system for maximizing the cost efficiency and benefits of adsorbents [60]. Figure 7d illustrates the effects of adsorbent dosage on the MB adsorption capacity and removal percentage. When 0.1 g of the nanocomposite beads was applied, the MB adsorption capacity was 50 mg/g, constituting a removal percentage of 54%. With increasing beads dosage above 0.1 g, the adsorption capacity decreased, but the dye removal percentage increased gradually, eventually reaching an equilibrium at a maximum value of 94.64% on using beads dosage of 1.5 g. This is because the concentration of dye in the solution was fixed, so excessive adsorbent dosage caused the active sites of the adsorbent to compete, resulting in a decrease in the amount of dye adsorbed per unit surface area [59,60]. Accordingly, the optimal beads dosage for MB removal was set as 1.5 g.

#### 3.5.5. Effect of Temperature

The influence of temperature on the removal rate is an important factor to investigate dye adsorption in industrial wastewater at different temperatures [55,58]. The dependence of the MB removal percentage on the temperature was examined at various temperatures ranging from 20–50 °C as presented in Figure 7e. Obviously, the removal percentage increased as the temperature rises from 20 to 30 °C. More specifically, the removal percentage reached 95% after 200 min at 30 °C, while it took 300 min to reach 90% at 20 °C. However, the removal percentage of MB at 40 °C and 50 °C revealed insignificant improvement compared with that at 30 °C. This finding was attributed to the increased mobility of the MB molecules at high temperature, which enhanced the MB molecules quickly get into the interior of hydrogel beads, resulting in an improvement of the removal percentage [57,61]. This behavior confirmed that the adsorption of dyes has an endothermic nature [61]. Hence, 30 °C was chosen as the optimum solution temperature.

#### 3.5.6. Reusability of Adsorbent

For acceptance of a system at industrial scale, the system should be economically feasible and repetitive. The reusability performance of adsorbent determines the feasibility of large-scale practical application [48,49]. The prepared semi-IPN PVA–Alg/Bent nanocomposite beads can be quickly separated from solution by simple separation technology, which provides the possibility for subsequent desorption and reuse. The nanocomposite beads for the dyes are reused for six cycles, after washing with 0.1 M HCl solution as the eluent. The reutilization was determined by the sorption-desorption cycles of the beads. As shown in Figure 7f, the beads had good adsorption effect on MB after one cycle of adsorption-desorption tests, with a removal percentage of 94.32%. As the number of cycles increases, the removal percentage can be maintained at a steady value of about 90%, indicating that the adsorbent shows good structural stability [50]. The obtained results suggested that the nanocomposite beads have good application prospects because of the good adsorption performance, efficient reusability and low cost from using cheap, environment-friendly raw materials and a simple manufacturing process. Hence, the results mentioned above indicate the possibility of using the semi-IPN PVA–Alg/Bent nanocomposites hydrogel beads at an industrial scale for the removal of MB from industrial wastewaters.

### 3.6. Adsorption Isotherm Study

The isotherm models of Langmuir and Freundlich were employed to get more insights into the removal process of MB dye by semi-IPN PVA–Alg/Bent nanocomposite hydrogel beads. The Langmuir isotherm model is assumed to indicate affinity for homogenous monolayer adsorption, while the Freundlich isotherm model is used to recognize the adsorption of multiple adsorption layer surfaces [50,51]. Figure 8 displays the adsorption isotherms of the MB removal by the nanocomposite hydrogel beads. In the present study, both the Langmuir and Freundlich models showed very high R^2^ values of 0.9932 and 0.9968, respectively, which indicated the suitability of both models to describe the adsorption of MB on hydrogel beads [61]. Accordingly, the adsorption process was likely to comprises both monolayer adsorption on homogeneous surface and multilayer adsorption on heterogeneous surface. The results shown that, R_L_ values lay between 0.005 to 0.03 (0 < *R_L_*< 1) and 1/n was about 0.57 (0 < 1/*n* < 1) as well as *q_m_* of 51.34 mg/g that confirmed the feasibility and favorability of the MB adsorption on the surface of the beads [62,63]. A comparison of *q*_m_ value for MB removal of the PVA–Alg/Bent nanocomposite hydrogel beads with other adsorbents is listed in Table 1, which clearly indicate the excellent adsorption capacity of the designed beads.

### 3.7. Adsorption Kinetics Study

The kinetics of adsorption are an important index in defining the efficiency of sorption [58]. In this study, the kinetic behavior of MB sorption onto the beads was investigated using both the pseudo-first-order and pseudo-second-order kinetic models. Linear plots of *ln*(*q_e_* − *q_t_*) and *t*/*q_t_* versus contact time are shown in Figure 9a,b, respectively. It can be seen that the R^2^ value (0.9974) of the pseudo-second-order kinetic model was higher than that of the pseudo-first-order kinetic model (0.9615), hence, the pseudo-second-order kinetic model is more suitable for describing the adsorption behavior of MB onto the beads, and the main adsorption process of MB was dominated by a chemisorption mechanism. This result is in accord with previous reports on dye adsorption on polysaccharide-based adsorbents [64].

### 3.8. Adsorption Mechanism of Cationic MB Dye

Figure 10 displays an illustrative scheme for the proposed adsorption mechanism. As mentioned before, at alkaline pH values the negative surface charge of the PVA–Alg/Bent nanocomposite hydrogel gel adsorbent is increased due to the ionization of -COOH and –SiOH groups into –COO^−^ and –SiO^−^, respectively. Thus, the removal of MB by the designed adsorbent is due to the electrostatic interactions between the positive charges on the iminium groups (=N+) present in the MB dye molecules and the abundant negative charges of the nanocomposite adsorbent. In addition, hydrogen bonding occurs between the –OH groups present in the adsorbent surface and nitrogen atom of MB [61,73].

## 4. Conclusions

In the present study, a semi-IPN PVA–Alg/Bent nanocomposite hydrogel was successfully developed as efficient adsorbent for the removal of MB dye from aqueous solution. The designed adsorbent beads possesses a porous structure, high specific surface area and net negative surface charge and provide many channels for the rapid adsorption of MB dye. Adsorbent beads had a good stability and reusability after six repeated cycles of adsorption and desorption. Moreover, the adsorbent beads showed outstanding MB removal efficiency with easy handling and extraction after adsorption, as well as eco-friendliness and low cost. Therefore, they could be suitable for the removal of other toxic cationic dyes and harmful pollutants from wastewater at an industrial scale.

## Figures and Tables

**Figure 1 polymers-13-04000-f001:**
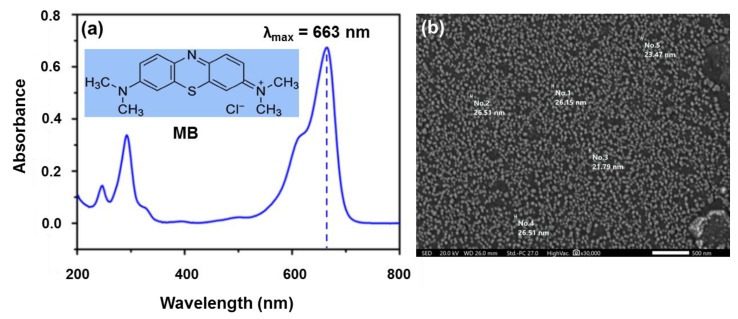
(**a**) UV/Vis spectrum of MB dye (the inset shows the chemical structure of MB) and (**b**) SEM micrograph of bentonite clay nanoparticles.

**Figure 2 polymers-13-04000-f002:**
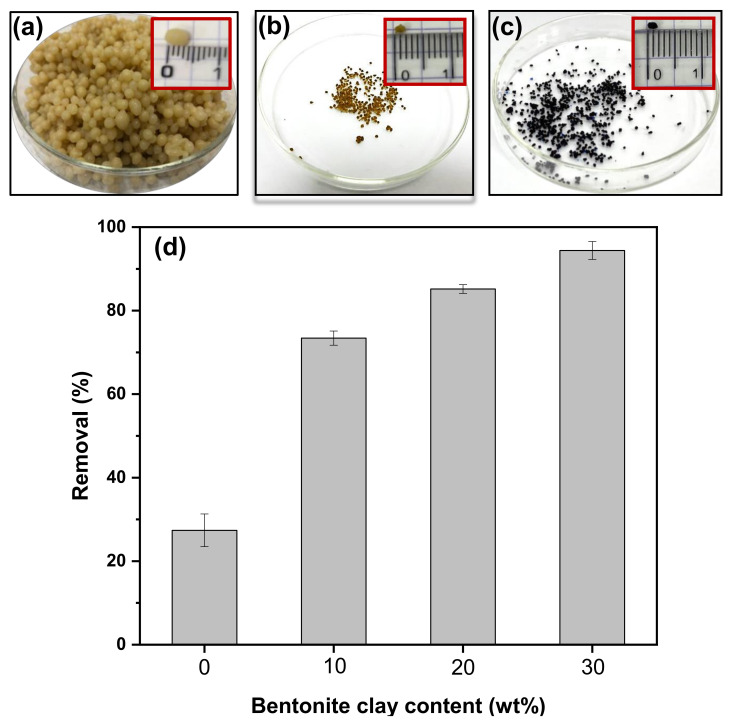
Semi-IPN PVA‒Alg/Bent nanocomposite hydrogel beads (**a**) before drying (inset shows the size of wet beads), (**b**) after 48 h air drying (inset shows the size of dried beads), (**c**) after adsorption of MB dye (inset shows the size of MB-adsorbed beads), and (**d**) effect of bentonite content (0, 10, 20, 30 wt%) on the removal percentage of MB (C_o_[MB] = 200 mg/L, V = 50 mL, pH = 8, beads dosage = 1.5 g, t = 300 min, T = 30 °C).

**Figure 3 polymers-13-04000-f003:**
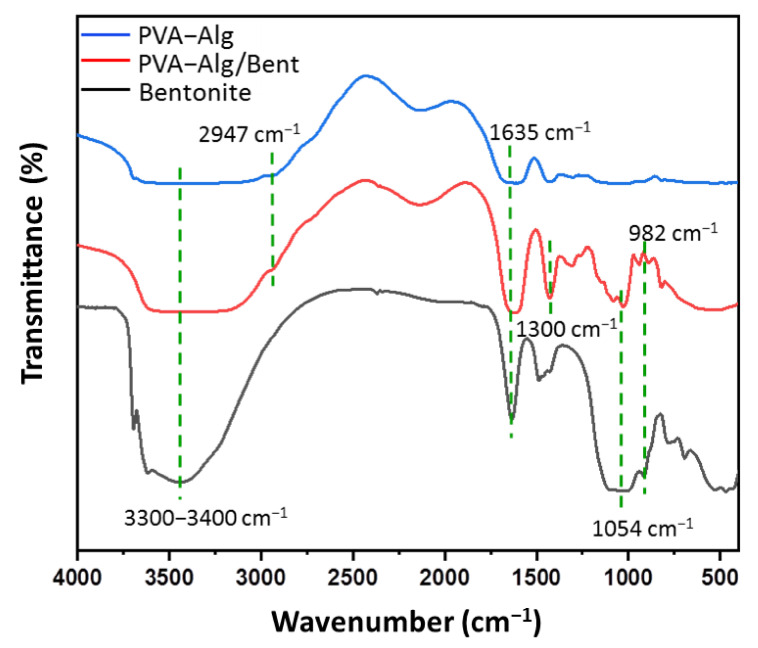
FTIR spectra of pristine Bent, semi-IPN PVA‒Alg hydrogel and semi-IPN PVA‒Alg/Bent nanocomposite hydrogel.

**Figure 4 polymers-13-04000-f004:**
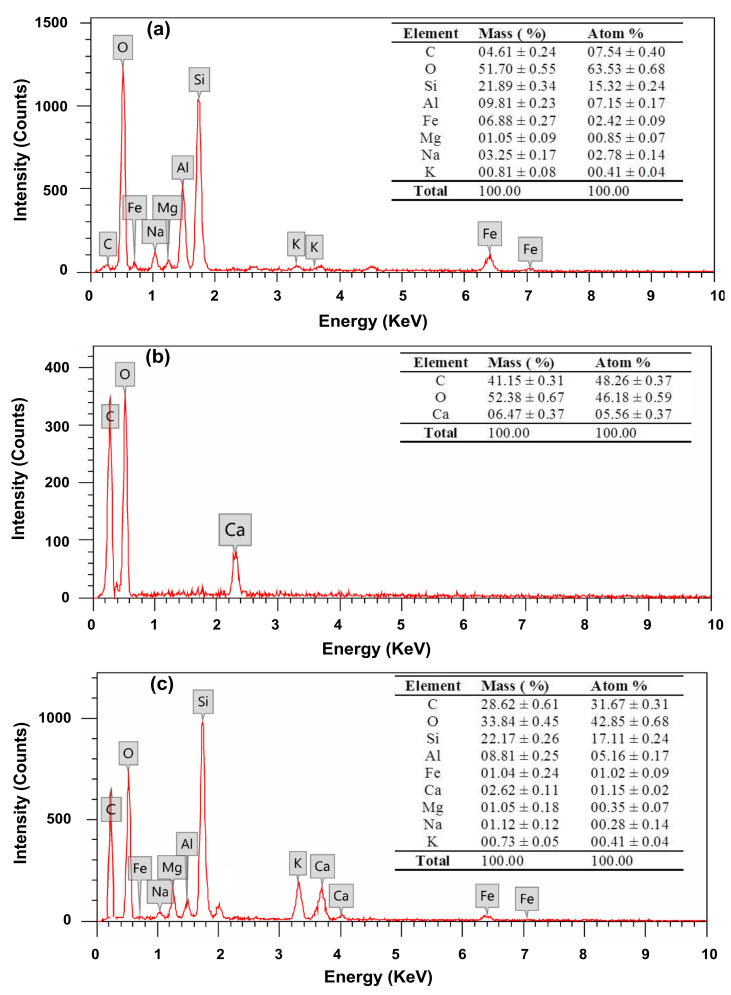
EDX spectra of (**a**) pristine bentonite clay (**b**) semi-IPN PVA‒Alg control hydrogel, and (**c**) semi-IPN PVA‒Alg/Bent nanocomposite hydrogel. The insets show the corresponding weight and atomic percentages of the elements present in the samples.

**Figure 5 polymers-13-04000-f005:**
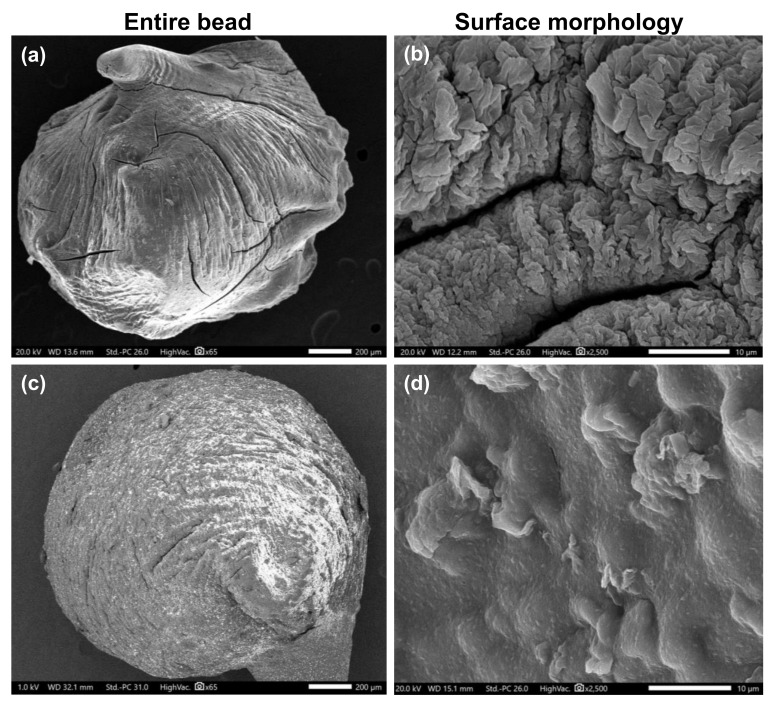
SEM micrographs of (**a**,**b**) semi-IPN PVA‒Alg control hydrogel beads, and (**c**,**d**) semi-IPN PVA‒Alg/Bent nanocomposite hydrogel beads.

**Figure 6 polymers-13-04000-f006:**
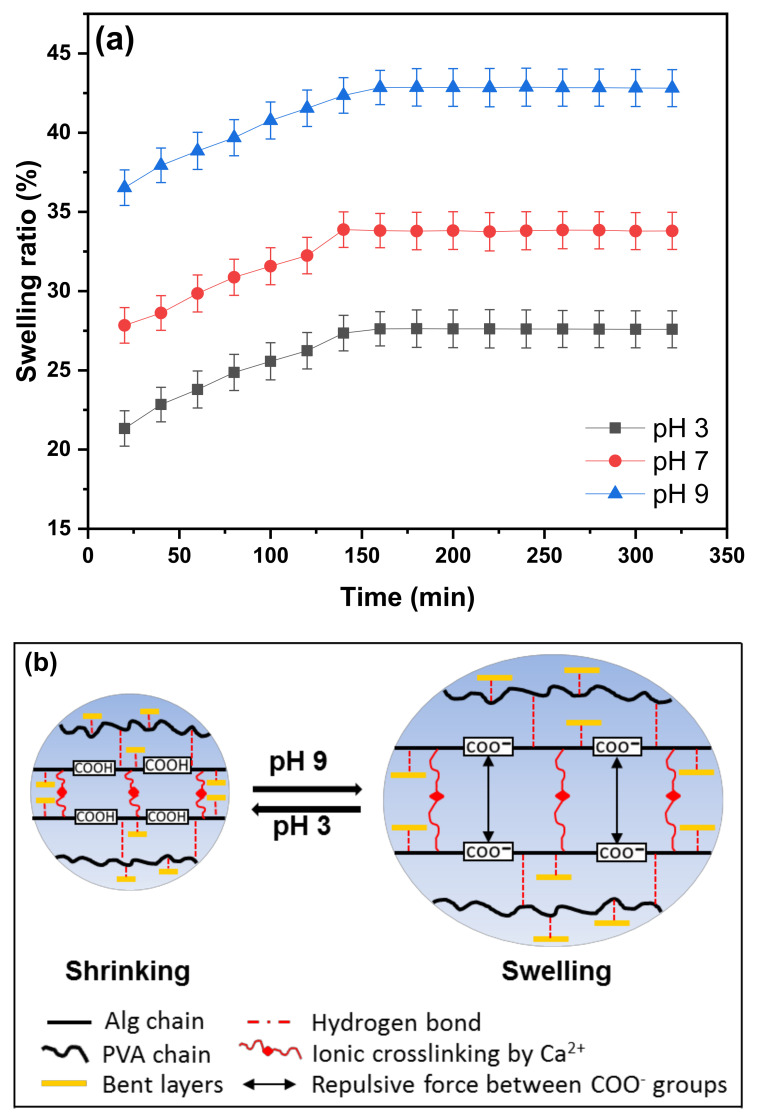
Swelling behavior of the semi-IPN PVA‒Alg/Bent nanocomposite hydrogel beads at different pH values: (**a**) swelling ratio and (**b**) Schematic illustration.

**Figure 7 polymers-13-04000-f007:**
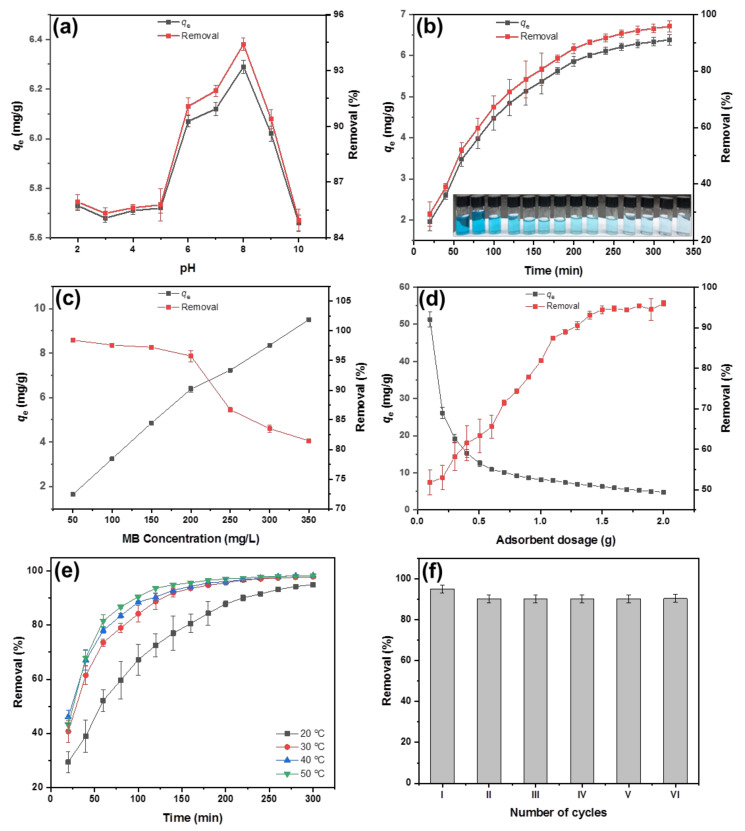
Effects of different factors on the adsorption capacity and removal percentage of MB onto the semi-IPN PVA‒Alg/Bent nanocomposite hydrogel beads: (**a**) pH (C_o_[MB] = 200 mg/L, V = 50 mL, beads dosage = 1.5 g, pH = 8, t = 300 min, T = 30 °C), (**b**) contact time (C_o_[MB] = 200 mg/L, V = 50 mL, pH = 8, beads dosage = 1.5 g, T = 30 °C), (**c**) MB initial concentration (V = 50 mL, pH = 8, beads dosage = 1.5 g, t = 300 min, T = 30 °C), (**d**) adsorbent dosage (C_o_[MB] = 200 mg/L, V = 50 mL, pH = 8, t = 300 min, T = 30 °C), (**e**) temperature (C_o_[MB] = 200 mg/L, V = 50 mL, pH = 8, beads dosage = 1.5 g, t = 300 min.), and (**f**) reusability (C_o_[MB] = 200 mg/L, V = 50 mL, pH = 8, beads dosage = 1.5 g, t = 300 min, T = 30 °C).

**Figure 8 polymers-13-04000-f008:**
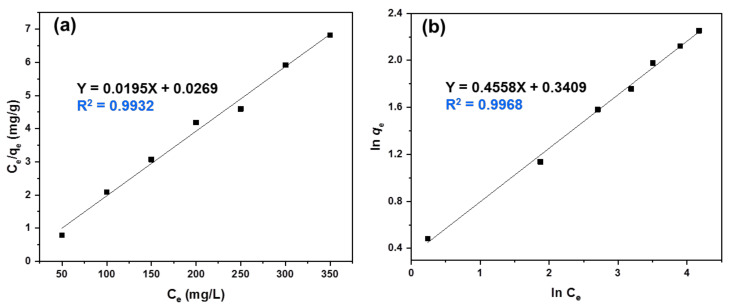
(**a**) The Langmuir adsorption isotherm and (**b**) Freundlich adsorption isotherm, for the adsorption of the MB onto the semi-IPN PVA‒Alg/Bent nanocomposite hydrogel beads.

**Figure 9 polymers-13-04000-f009:**
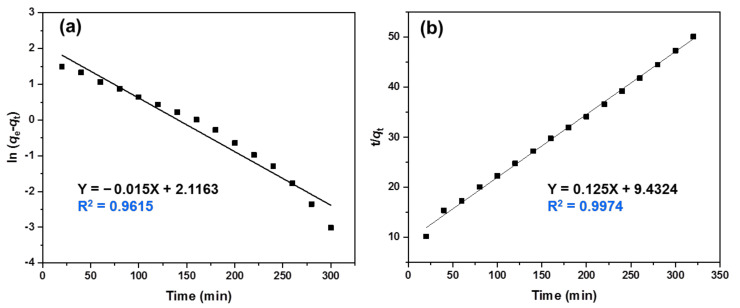
(**a**) The pseudo-first-order and (**b**) pseudo-second-order kinetic models for the adsorption of the MB onto the semi-IPN PVA‒Alg/Bent nanocomposite hydrogel beads.

**Figure 10 polymers-13-04000-f010:**
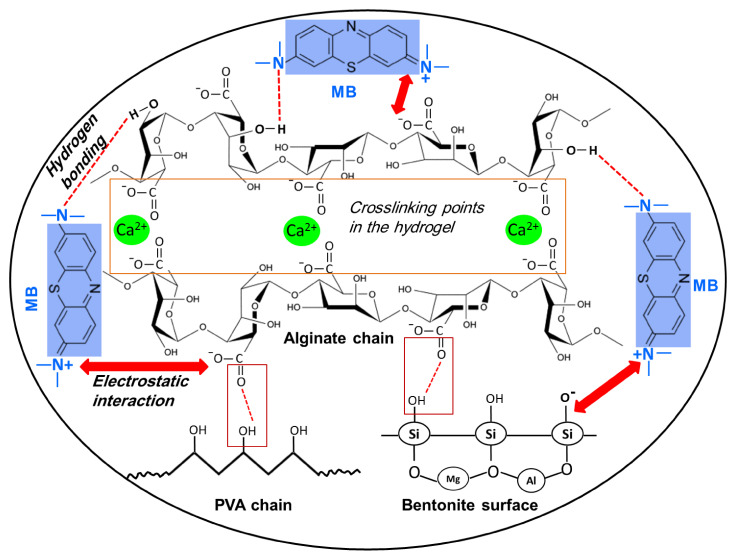
Proposed scheme indicating the possible interaction mechanism of the MB onto the PVA‒Alg/Bent nanocomposite hydrogel beads.

**Table 1 polymers-13-04000-t001:** Comparison of the designed semi-IPN PVA–Alg/Bent nanocomposite hydrogel beads with previously reported adsorbents for MB removal.

Adsorbent	Maximum Adsorption Capacity (mg/g)	References
Activated charcoal/β-cyclodextrin/Alg hydrogel beads	10.63	[49]
PVA/cyclodextrin-modified poly(acrylic acid) hydrogel	23.02	[58]
Carboxymethyl cellulose (CMC)-based hydrogel	25.00	[65]
CMC/k-carrageenan/montmorillonite (MMT) beads	12.50	[66]
Graphene oxide (GO)/Fe_3_O_4_/chitosan nanocomposite	30.10	[67]
PVA/CMC/halloysite nanoclay membrane	40.60	[68]
Corn stalk/MMT composite hydrogel	49.01	[69]
GO/Fe_3_O_4_/Alg nanocomposite	37.04	[70]
Chitosan/sepiolite composite	40.98	[71]
PVA/Bent hydrogel	27.90	[72]
PVA–Alg/Bent nanocomposite hydrogel beads	51.37	This study

## Data Availability

Not applicable.

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
