# Peer review of "Environmentally Friendly Polyvinyl Alcohol−Alginate/Bentonite Semi-Interpenetrating Polymer Network Nanocomposite Hydrogel Beads as an Efficient Adsorbent for the Removal of Methylene Blue from Aqueous Solution"

_polymers, 2021, doi:10.3390/polym13224000_

Round 1
Reviewer 1 Report
The manuscript entitled, ‘Environmentally Friendly Polyvinyl alcohol-alginate/bentonite Nanocomposite Hydrogel Beads as Efficient Adsorbents for Removal of Toxic Methylene Blue from Aqueous Solution’ discussed PVA based hydrogels for dye removal applications. The review work is in quite detail but it needs to address some queries. I am mentioning some of them;
- The composite hydrogel is semi-interpenetrating in nature. Author should mention it in the manuscript. It will be better to discuss the semi-IPN type hydrogels in the introduction section.
- Author wrote ‘CaCl2 solution (3%, 600 mL)’. It will be more scientific if the author mention the concentration of CaCl2 in terms of molarity or wt% with respect to polymer or total mixture.
- Author wrote that the SEM was done at ‘25 keV’; first of all it is ‘kV’, not ‘keV’; and the voltage is too high for hydrogel sample. Author is requested to double check this voltage.
- It will be better if the author put one table comparing to other’s work and their own work.
- There are some recent articles which could improve the literature review of the manuscript. I am recommending some of them: Ultrasonics sonochemistry60 (2020): 104797; Polymer-Plastics Technology and Engineering7 (2017): 744-761; Environmental Research 195 (2021): 110809; Journal of Water Process Engineering 44 (2021): 102351.
- Scale bar should be provided in Figure 2(a-c).
- Figure 1b is needed to explain with size distribution/range of sizes of the Bent nanoparticles.
Reviewer 2 Report
This manuscript demonstrates the synthesis of Polyvinyl alcohol−alginate/bentonite Nanocomposite Hydrogel Beads for Efficiently Adsorbing for Methylene Blue. It can be accepted after a minor revision.
- Remove "toxic" from the title as MB exhibits little toxicity.
- Kinetics and isotherm study needs critical supports from previous reports, for example, Environmental Science & Technology 2021 55 (8), 4287-4304; ACS ES&T Engineering 2021 1 (4), 623-661; Journal of Cleaner Production 2019, 232, 774-783.
- Zeta potential of prepared adsorbent should be measured.
- Detailed reusability procedure of prepared adsorbent should be added.
Round 2
Reviewer 1 Report
It can be published in its present state.